# Characterization of the Interactions between Minocycline Hydrochloride and Trypsin with Spectroscopic and Molecular Docking Technology

**DOI:** 10.3390/molecules28062656

**Published:** 2023-03-15

**Authors:** Xiaoxia Wang, Jisheng Sun, Litong Ma, Zhihua Nie, Huazheng Sai, Jianguo Cheng, Jianguo Duan

**Affiliations:** 1School of Chemistry and Chemical Engineering, Inner Mongolia University of Science and Technology, Baotou 014010, China; 2Inner Mongolia Engineering Research Center of Comprehensive Utilization of Bio-coal Chemical Industry, Baotou 014010, China; 3School of Life Sciences, Tsinghua University, Beijing 100084, China

**Keywords:** minocycline hydrochloride, trypsin, multispectral analysis, molecular docking simulation method

## Abstract

In the current study, the interaction of minocycline hydrochloride (MC) and trypsin (TRP) was studied using fluorescence spectroscopy, synchronous fluorescence spectroscopy, three-dimensional fluorescence spectroscopy, UV-Vis spectroscopy, and molecular docking simulation techniques. The results show that the fluorescence quenching of trypsin at different degrees can be caused by minocycline hydrochloride at different temperatures. According to the Stern-Volmer equation, the fluorescence quenching type was static quenching. By calculating critical distance, we concluded that there is a possibility of non-radiative energy transfer between minocycline hydrochloride and trypsin. The effect of minocycline hydrochloride on the secondary structure of trypsin was demonstrated using ultraviolet spectroscopy. Synchronous fluorescence spectroscopy showed that minocycline hydrochloride could bind to tryptophan residues in trypsin, resulting in corresponding changes in the secondary structure of trypsin. Three-dimensional fluorescence spectroscopy showed that minocycline hydrochloride had a particular effect on the microenvironment of trypsin that led to changes in the secondary structure of trypsin. The molecular docking technique demonstrated that the binding of minocycline hydrochloride and trypsin was stable. Circular dichroism showed that the secondary structure of trypsin could be changed by minocycline hydrochloride.

## 1. Introduction

Minocycline hydrochloride is a broad-spectrum antibiotic, which usually exists in the form of hydrochloride. Minocycline hydrochloride has a good bactericidal effect [1], high efficiency, and long-lasting medicinal characteristics. Minocycline hydrochloride is listed as a semi-synthetic tetracycline broad-spectrum antibiotic, and it has the strongest antibacterial activity among tetracycline antibiotics [2]. Its antibacterial spectrum is almost the same as that of tetracycline. In contrast, trypsin is a kind of protease. Trypsin is mostly found in vertebrates, where it acts as a digestive enzyme [3,4,5], but it is also found in organisms such as silkworms, asterid, squirts, and actinomycetes. Trypsin has reduced inflammation and swelling in clinical studies [6], and it is used in leather manufacturing, raw silk processing, food processing, and so on [7].

In recent years, studying the interaction between small drug molecules and trypsin using fluorescence spectra has become an important topic in chemical research; fluorescence spectra helps explore mysteries in the life science field [8]. With the gradual improvement of modern science and technology, researchers have used a variety of methods to investigate the reaction between small drug molecules and trypsin from different perspectives. Momeni Lida, Shareghi Behzad [9] et al. studied the interaction between trypsin and 2-propanol using molecular simulation and spectroscopy. Meti Manjunath D [10] et al. used spectroscopy and CD techniques to study the inhibitory effects of drugs on trypsin. Sahin Selmihan, Calapoglu Furkan [11] et al. investigated binding interaction behavior between antiemetic drugs and trypsin using spectroscopy and molecular docking. However, as yet no study in the literature has reported on minocycline and trypsin. In this experiment, through the simulation of physiological conditions, fluorescence spectroscopy was used to determine the quenching type, mechanism of action, binding constant, and the number of binding sites [12]. Circular dichroism spectrum, three-dimensional fluorescence spectrum, and ultraviolet absorption spectrum analyses were used to determine both the type and change of the secondary structure of the reaction [13]. Finally, the molecular mechanism of the interaction between minocycline and trypsin was identified. This investigation enriches the understanding of the mechanism by which minocycline hydrochloride binds to trypsin. It provides a reference for further research on the distribution, metabolism, and toxicological mechanisms of minocycline hydrochloride in vivo.

## 2. Results and Discussion

### 2.1. Fluorescence Quenching Spectra

Figure 1 shows the change in the fluorescence intensity of TRP as MC concentration increased at three temperatures (298 K, 303 K, 308 K). MC is a quencher. Under the excitation of TRP at 280 nm, the maximum emission wavelength of TRP was about 353 nm. As MC concentration increased, the fluorescence intensity of TRP steadily decreased. However, there was no obvious change in the position of the emission peak, which indicated that MC could cause the quenching reaction of TRP at different temperatures, and that the quenching effect was strong.

### 2.2. Determination of Fluorescence Quenching Mechanism

Static and dynamic quenching are two types of fluorescence quenching in reactions between drug molecules and protein macromolecules [14]. Dynamic quenching occurs after the quenching agent collides with the excited state of the fluorescent substance. The interaction between quenchers and ground-state molecules of fluorescent substances, and the formation of complexes, leads to static quenching. In dynamic quenching, the number of ion collisions in the system increases with system temperature, and the free electrons also increase the rate and frequency of ion transfer in the system, so that constant quenching increases. Otherwise, static quenching occurs [15,16].

In general, the dynamic quenching reaction is caused by the thermal motion of the collision reaction between drugs and protein molecules. It is usually expressed by the Stern-Volmer equation [17]:(1)$$F_{0}/F=1+K_{q}\tau_{0}\left[Q\right]=1+K_{SV}\left[Q\right]$$
where $F_{0}$ corresponds to the fluorescence intensity of MC in TRP without a quenching agent, F corresponds to the fluorescence intensity of TRP in MC with a quenching agent, the maximum dynamic fluorescence rate constant of the reaction of fluorescence quenchers (drugs) with proteins is generally about 2 × 10^10^ L/(mol·s),$\tau_{0}$ corresponds to the endogenous fluorescence lifetime of biological macromolecules (about 10^−8^ s), $\left[Q\right]$ corresponds to the final concentration of MC solution in the experiment, $K_{q}$ corresponds to the dynamic fluorescence quenching rate constant, and $K_{SV}$ corresponds to the dynamic quenching constant.

The fluorescence quenching of TRP by MC was assumed to be dynamic quenching. With MC concentration as the horizontal coordinate and $F_{0}/F$ as the vertical coordinate, Stern-Volmer curves of MC-TRP at three temperatures (298 K, 303 K, and 308 K) were drawn (as shown in Figure 2).

Table 1 shows that the size of $K_{q}$ at each temperature was close to 10^13^, which was much larger than the maximum collision quenching rate constant. This indicated that the fluorescence quenching interaction between MC and TRP was static quenching. As temperature increased, the quenching constant $K_{SV}$ decreased, demonstrating that the type of interaction between TRP and MC was static quenching.

### 2.3. The Binding Constant and Number of Binding Sites (n)

According to the above conclusions, TRP and MC exhibited static quenching. When MC reacts with TRP, MC has n binding sites on TRP, which can be expressed by the Scatchard equation [18,19]:(2)$$\mathrm{lg}\left[{(F}_{0}-F)/F\right]=\mathrm{lg}K_{A}+n\mathrm{lg}\left[Q\right]$$
where $K_{A}$ corresponds to the binding constant between TRP and MC, and n corresponds to the number of binding sites between TRP and MC. lg⁡[Q] was used as the horizontal coordinate, and $\mathrm{lg}\left[{(F}_{0}-F)/F\right]$ was used as the vertical coordinate to make a double logarithmic plot (Figure 3). Binding-site number is calculated using the slope of the curve. The constant K_A_ is calculated according to the intercept of the curve (as shown in Figure 3).

It can be seen in Table 2 that the binding constants of MC and TRP at different temperatures were all greater than 10^4^, indicating that they bind closely and can be transported and stored well in vivo. Additionally, the number of binding sites was close to 1, indicating that MC and TRP were combined at 1:1. As temperature increased, the value of the binding constant $K_{A}$ decreased, which further demonstrated that the reaction between MC and TRP is static quenching.

### 2.4. Thermodynamic Parameters and Main Forces of the Interaction between TRP and MC

In most cases, interactions between biological macromolecules (proteins) and drug molecules are hydrophobic forces, van der Waals forces, hydrogen bonds, and electrostatic forces [20]. Usually, the type of interaction between protein and drug can be judged by Ross theory [21]: when ∆H > 0, ∆S > 0, the force type between protein and drug is hydrophobic force; when ∆H < 0, ∆S < 0, the force types between protein and drug are hydrogen bond and van der Waals force; when ∆H < 0, ∆S > 0, the force type between protein and drug is electrostatic force. In general, the thermodynamic parameters of MC and TRP at different temperatures can be calculated by using van’t Hoff equations.
(3)∆G=∆H−T∆S
(4)$$∆G=-RT\ln K_{A}$$
(5)$$\ln\left(K_{2}-K_{1}\right)=∆H\left(1/T_{1}-1/T_{2}\right)/R$$
where $K_{A}$ corresponds to the binding constant at the corresponding temperature, and R corresponds to the molar gas constant, its value is approximately 8.314 J/(mol·K). ∆H, ∆S, and ∆G represent the change in enthalpy, entropy, and Gibbs energy, respectively. T is the thermodynamic temperature during the experiment. The van’t Hoff equation was used to calculate the thermodynamic parameters (∆H, ∆S, and ∆G), as shown in Table 3.

According to Table 3, ∆H < 0 and ∆S < 0, from which it can be inferred that the main interaction forms between MC and TRP were van der Waals force and hydrogen bond. ∆G < 0 indicates that the static quenching of TRP by MC is a spontaneous and exothermic binding process.

### 2.5. The Combination Distance between MC and TRP

According to the Förster dipole-dipole non-radiative energy transfer mechanism [22,23], non-radiative energy transfer occurs between an energy donor and an energy receiver when they meet the following conditions simultaneously: if the concentrations of the recipient and the donor are very close, the fluorescence spectrum of the donor overlaps with the UV absorption spectrum of the recipient, and the overlap distance is less than 7 nm. The critical distance (r) between protein and drug, the fluorescence emission spectrum of energy donor (TRP), and the overlap integral of UV-visible absorption spectrum of energy receptor (MC), and the critical energy transfer distance R_0_ are calculated using the following formula:(6)$$E=1-F/F_{0}=R_{0}^{6}/\left(R_{0}^{6}+r^{6}\right)$$
(7)$$R_{0}^{6}=8.8\times 10^{-25}K^{2}N^{-4}\phi J$$
(8)$$J=\sum F\left(\lambda \right)\varepsilon \left(\lambda \right)\lambda^{4}∆\lambda /\sum F\left(\lambda \right)∆\lambda$$
where $F_{0}$ is the fluorescence intensity of TRP without MC, F is the intensity of the fluorescence emission peak when the concentration ratio of MC to TRP is 1:1, $K^{2}$ is the average value of the randomly distributed orientation factor of energy recipient MC and energy donor TRP, ϕ is the fluorescence quantum yield of energy donor TRP (ϕ = 0.118 J), and N is the refractive index of the medium (the average value of water and organic matter N = 1.336). $\varepsilon \left(\lambda \right)$ is the molar absorptivity of energy acceptor MC at wavelength λ, and $F\left(\lambda \right)$ is the fluorescence intensity of energy donor MC at wavelength λ.

It can be seen in Figure 4 that there was a certain overlap between the UV-visible absorption spectrum curve of MC and the fluorescence spectrum curve of TRP; this shows that there was a non-radiative energy transfer between MC and TRP. Using matrix division and integration, the overlapping integral value of MC and TRP (concentration ratio is 1:1) was calculated to be J = 6.083 × 10^−14^ cm^3^ ·mol/L $R_{0}$ = 3.313 nm, r = 4.379 nm. The results showed that the critical distance between MC and TRP was less than 7 nm, which demonstrated that the fluorescence quenching of MC and TRP is caused by non-radiative energy transfer.

### 2.6. Molecular Docking Study

Molecular docking simulation technology is an advanced method used to study the binding sites of drugs and proteins [24]. Molecular docking simulation technology can be used for a more detailed study of the binding sites of MC and TRP, the interaction between them, specific distances, and amino acid residues that contribute to the binding of TRP [25]. The molecular docking of the MC and TRP was simulated by DS2016client.exe software, and the result is shown in Figure 5.

It can be seen in Figure 5 that the interaction forces between the amino acid residues on MC and TRP were mainly hydrogen bond, van der Waals force, and unfavorable bump, and there were a few hydrophobic forces (in the form of Pi-Sulfur and Pi-Alkyl). Among them, the interaction forces between MC and amino acid residues CYS700, HIS682, and TYR681 in TRP were hydrogen bond, and the bond lengths were 4.07 Å, 5.36 Å, and 4.9 Å, respectively. The interaction force between MC and amino acid residues HIS699, SER836, GLY834, GLN833, LYS702, and TYR701 were van der Waals force. The interaction force between amino acid residue CYS684 and MC was a Pi-Sulfur bond, and the bond length was 5.50 Å. The interaction force between amino acid residue PHE683 and MC was Pi-Alkyl, with a bond length of 3.88 Å. In addition, the binding force of amino acid residues PHE683 and TYR681 with MC were unfavorable collision, and the bond lengths were 3.62 Å and 2.93 Å, respectively. This demonstrated that all kinds of forces stabilize MC and TRP binding.

### 2.7. Effect of MC on the Secondary Structure of TRP

#### 2.7.1. Synchronous Fluorescence Spectral Analysis of the Interaction between MC and TRP

The presence of three amino acid residues in TRP (tryptophan residue (Trp), tyrosine residue (Tyr), and phenylalanine residue (phe)) makes TRP emit strong fluorescence at specific wavelengths [26]. The quantum yield of phenylalanine is relatively low. The fluorescence intensity of tyrosine and tryptophan decreased with increased drug concentration. In the synchronous fluorescence spectrum, the excitation wavelength and emission wavelength of the instrument are set to a fixed value. Then the synchronous fluorescence spectrum is generated by the scanning excitation and emission monochromator. When Δ λ = 15 nm, it generated the fluorescence spectrum of the tyrosine residue. When Δ λ = 60 nm, it generated the fluorescence spectrum of tryptophan residue. The maximum emission wavelength of amino acid residues in protein are usually related to the polarity and hydrophobicity of their microenvironments, so the structural change in protein can be judged by the change in amino acid residue wavelength.

The synchronous fluorescence spectra of TRP-MC with wavelength difference Δ λ = 15 nm and Δ λ = 60 nm are shown in Figure 6.

It can be seen in Figure 6 that when the concentration of TRP was fixed, and the concentration of MC gradually increased, the emission peak intensities of tyrosine and tryptophan residues in TRP were quenched to varying degrees. When Δ λ = 15 nm, the position of the maximum peak showed almost no change, indicating that the drug had no obvious quenching effect on tyrosine residues. When Δ λ = 60 nm, the position of the maximum peak shifted from 280 nm to 287 nm, the redshift was 7 nm, and the quenching degree was strong, indicating that MC can change the microenvironment of tryptophan residues in TRP, increase the polarity of tryptophan residues, decrease their hydrophobicity, and increase their hydrophilicity. Additionally, MC can change the secondary structure of TRP, and it is mainly bound to the tryptophan residues of TRP.

#### 2.7.2. UV Spectroscopic Analysis of the Interaction between MC and TRP

Ultraviolet spectroscopy is a common method to study the change in the secondary structure of a protein when it reacts with drugs at different concentrations. Using secondary distilled water as a reference, the ultraviolet absorption spectrum of the interaction between MC and TRP was determined, and is shown in Figure 7.

It can be seen in Figure 7 that there was a maximum absorption peak at 280 nm. As MC concentration increased, the position of the highest absorption peak of TRP had a red shift from 280 nm to 285 nm. The results showed that the binding of MC to the base pair of TRP resulted in a π-π * transition of amino acid residues in TRP, which reduced the energy and changed the conformation of the protein [27]. The absorbance of the maximum absorption peak of TRP increased noticeably, and only a static quenching absorption spectrum changes with a change of ground state molecule. Therefore, the ultraviolet absorption spectrum further demonstrated that the type of interaction between MC and TRP is static quenching. It also shows that MC can affect the secondary structure of TRP.

#### 2.7.3. Three-Dimensional Fluorescence Spectral Analysis of MC and TRP

Three-dimensional fluorescence spectroscopy is generally used to study the conformational changes of proteins after reactions between drugs and proteins. The three-dimensional fluorescence spectrogram can obtain the required information more intuitively than the two-dimensional spectrogram. A three-dimensional fluorescence spectrogram is generally represented by a three-dimensional projection map and a contour map [28]. To further study the effect of MC on the secondary structure of TRP, the three-dimensional fluorescence spectra of TRP and MC-TRP systems were measured. The three-dimensional fluorescence spectra and contour maps were made, as shown in Figure 8 and Figure 9. The relevant parameters are listed in Table 4.

Figure 7 and Figure 8 showed that both peak a and peak b were in the shape of a “ridge”, called a Rayleigh scattering peak (*λ_ex_
*= *λ_em_*). Peak 1 and peak 2 were “hump” shapes, which were typical fluorescence peaks (2*λ_ex_
*= *λ_em_*). The characteristic parameters (*λ_ex_/λ_em_*, *F*) of the two fluorescence peaks were measured: (285/353, 414.3) for peak 1, and (220/349, 815.5) for peak 2. After adding MC, the characteristic parameters (*λ_ex_/λ_em_*, *F*) of the two fluorescence peaks of MC-TRP were (280/354, 292.8) for peak 1, and (220/354, 598.1) for peak 2.

Compared with A and B systems, it was found that the fluorescence intensity of peak-1 decreased from 414.3 to 292.8, and the peak of the maximum emission wavelength was red-shifted by 1 nm. The fluorescence intensity of peak 2 decreased noticeably from 815.5 to 598.1, and the maximum emission peak was red-shifted by 5 nm. After the addition of MC, the intensity of the Rayleigh scattering peak and the fluorescence peak of TRP decreased. This indicated that the protective water film on the surface of the protein was destroyed after the surface of TRP was bound to MC, so the protein became more dispersed and the fluorescence intensity of the protein decreased. The results showed that MC interacted with TRP, and also that MC had a particular effect on the microenvironment around TRP, reducing the polarity of the environment, enhancing the hydrophobicity, and changing the secondary structure of the peptide chain of TRP.

#### 2.7.4. Circular Dichroism Analysis of MC by TRP

Circular dichroism is a special method used to study the interactions between drugs and proteins [29]. To further study the effect of MC on the conformation of TRP, circular dichroism was used to determine the MC-TRP system. As shown in Figure 10, TRP had a single negative peak at 208 nm, and the intensity of the negative peak decreased noticeably after adding MC. The circular dichroism data were analyzed and calculated by the software, and the results are shown in Table 5.

It can be seen in Table 5 that the content of the β-sheet played a dominant role in the interaction between MC and TRP. After adding MC, the content of α-Helix increased from 10.8% to 12.6%, the content of β-sheet increased from 50.5% to 53.2%, the content of β-turn decreased from 19.2% to 18.9%, and the content of random coil decreased from 27% to 25.8%. This showed that MC changed the secondary structure of TRP.

## 3. Experimental Section

### 3.1. Instruments

Fluorescence spectrophotometer (LS-55, PerkinElmer company, Waltham, MA, USA); UV-Vis spectrophotometer (CARY5000, Agilent Technologies, Los Angeles, CA, USA); digital display constant temperature water bath pot (SHA-B, Jintan Medical instrument Factory, Nanjing, China); electronic balance (TP-114, Denver Instrument Beijing Co Limited, Beijing, China); PH meter (EL20, METTLER TOLEDO, Shanghai, China); ultrasonic cleaner (DL-120E, Shanghai Zhixin Instrument Co., Ltd., Shanghai, China).

### 3.2. Reagents

Tris (hydroxymethyl) aminomethane (Tris, purity ≥ 99%), Minocycline (MC, purity ≥ 99%), Trypsin (TRP, purity ≥ 99%), and sodium chloride were obtained from China Food Research Institute. The BSA reserve solutions (1 × 10^−4^ mol/L) were prepared with secondary distilled water. The Tris-HCl buffer solution (0.5 mol/L, pH 7.40) was prepared with secondary distilled water and HCl. The TRP reserve solutions (3 × 10^−5^ mol/L) were prepared with Tris-HCl buffer solution (0.5 mol/L, pH 7.40). After preparation, the configured BSA solution, Tris-HCL buffer solution, and TRP solution were stored in the refrigerator.

The reagents used in the experiment were analytical reagent grade, and the water used in the experiment was secondary distilled water without fluorescent impurities.

### 3.3. Determination of Fluorescence Spectra, Synchronous Spectra, and Three-Dimensional Fluorescence Spectra of TRP and MC

Ten identical color tubes (10 mL) were filled with 1 mL TRP (3 × 10^−5^ mol/L), and then various amounts of MC solution [(0,0.5, 1.0, 1.5, 2.0, 2.5, 3.0, 3.5, 4.0, 4.5) × 10^−5^ mol/L, respectively] were added. Next, secondary deionized water was added into the colorimetric tube to the 10 mL scale line, to ensure constant volume in each tube. After shaking, they were left for static use. Then the fluorescence spectrophotometer was set to fluorescence scanning mode, and the following parameters were set: the excitation wavelength was 280 nm, the slit width was 7 nm, the scanning speed was 1500 nm/min, and the fluorescence emission spectrum was recorded in the wavelength range of 290 nm–550 nm. The synchronous fluorescence spectra were scanned under Δ λ = 15 nm and Δ λ = 60 nm conditions. Three-dimensional fluorescence scanning was carried out with an excitation wavelength of 200 nm and an emission wavelength in the range of 200 nm–500 nm, and the scanning speed was 1500 nm/min. The parameters were stored and used for measurements at three temperatures (298k, 303K, 308K, respectively).

### 3.4. UV-Vis Absorption Spectrum of TRP and MC

Ten identical color tubes (10 mL) were filled with 1 mL TRP (3 × 10^−5^ mol/L), and then various amounts of MC solution were added [(0,0.5, 1.0, 1.5, 2.0, 2.5, 3.0, 3.5, 4.0, 4.5) × 10^−5^ mol/L, respectively]. Next, secondary deionized water was added into the colorimetric tube to the 10 mL scale line, to ensure constant volume in each tube. After shaking, they were left for static use. The UV absorption spectra of trypsin with different concentrations of minocycline were measured at the wavelength of 190 nm–400 nm.

### 3.5. Determination of Binding Distance of TRP and MC

The MC solution (0.3 × 10^−5^ mol/L) was determined by using the UV-Vis spectrophotometer, and the TRP (1.0 × 10^−4^ mol/L) and TPL-BSA mixture solutions (0.3 × 10^−5^ mol/L) were determined using the fluorescence spectrophotometer. The measuring wavelength of the UV-Vis spectrophotometer was between 190 nm and 400 nm. The equipment parameters of the fluorescence spectrophotometer were the same as in Section 2.3

### 3.6. Molecular Docking Simulation Technology of TRP and MC

The crystal structure of TRP was found in the protein database. The molecular structure formula of MC was drawn by ChemOffice software, and the obtained MC molecular structure formula was saved in pdf format. Next, it was converted into a 3D structure and saved. The binding sites of TRP and MC were selected by the DS2016client.exe program, and then molecular docking was carried out.

### 3.7. Circular Dichroism Spectrum

TRP (1 × 10^−5^ mol/L) was used as a reference solution. The concentration ratio of TRP (2 × 10^−5^ mol/L) to MC (5 × 10^−5^ mol/L) was 1:4. The scanning range of circular dichroism was 190 nm–260 nm, the width of the slit was 1 nm, and the scanning speed was 3.3 nm/s. All the data were scanned four times to obtain an average value.

## 4. Conclusions

In this paper, the fluorescence quenching spectra of TRP and MC at different temperatures showed that the fluorescence intensity of TRP decreased gradually as MC concentration increased, which indicated that MC caused the quenching reaction of TRP. According to the Stern-Volmer curve, linear equation, and correlation coefficient, the dynamic quenching constant K_SV_ decreased with the increase in temperature, indicating that static quenching occurred in the interaction between TRP and MC. By calculating both the binding constant and the number of binding sites between TRP and MC, it was found that the binding constant decreased as temperature increased, which was consistent with the temperature dependence of the quenching constant K_SV_. Additionally, the number of binding sites was close to 1, which further demonstrated that the reaction of TRP with MC was static quenching. The thermodynamic parameters of the reaction between TRP and MC were ΔH < 0, ΔS < 0, indicating that the forces between them were van der Waals force and hydrogen bond. As ΔG < 0, the interaction process was a spontaneous and exothermic process with decreased free energy (ΔH < 0). The overlap integral value of the critical distance was J = 6.083 × 10^−14^ cm^3^.mol/L, R_0_ = 3.313 nm, r = 4.379 nm, and the binding distance was r < 7 nm. It was demonstrated that the fluorescence quenching process was caused by non-radiative energy transfer. Additionally, the synchronous fluorescence spectra of TRP and MC showed that MC was mainly bound to tryptophan (Try) residue in TRP. MC changed the microenvironment of the tryptophan residue in TRP, which made the polarity of tryptophan residue stronger, decreased hydrophobicity, increased hydrophilicity, and changed the conformation of TRP. The ultraviolet fluorescence spectra of TRP and MC showed that the interaction between MC and TRP was static quenching, and showed that MC can affect the secondary structure of TRP. The three-dimensional fluorescence spectra of TRP and MC demonstrated that MC caused a change in the microenvironment of TRP, thus changing the conformation of TRP. The molecular docking simulation technique demonstrated that the various interactions between MC and TRP amino acid residues created stable binds between MC and TRP. Circular dichroism analysis of TRP and MC showed that trypsin was dominated by β-sheet folding, and that MC changed the secondary structure of TRP.

## Figures and Tables

**Figure 1 molecules-28-02656-f001:**
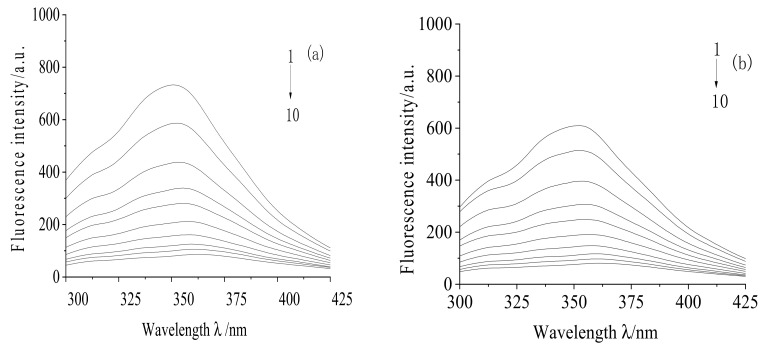
Fluorescence emission spectra in the MC-TPS interaction system under different temperatures. (**a**): 298 K, (**b**): 303 K, (**c**): 308 K; pH = 7.40. C_(TRP)_ = 3 × 10^−5^ mol/L; C_(MC)_ (1–10): 0, 0.5, 1, 1.5, 2, 2.5, 3, 3.5, 4, 4.5 × 10^−4^ mol/L.

**Figure 2 molecules-28-02656-f002:**
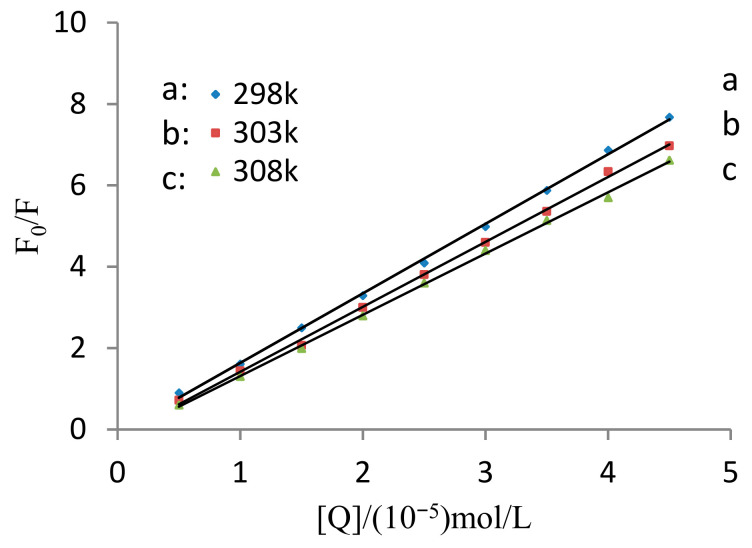
Stern-Volmer equation curve for the quenching of TRP by MC at different temperatures.

**Figure 3 molecules-28-02656-f003:**
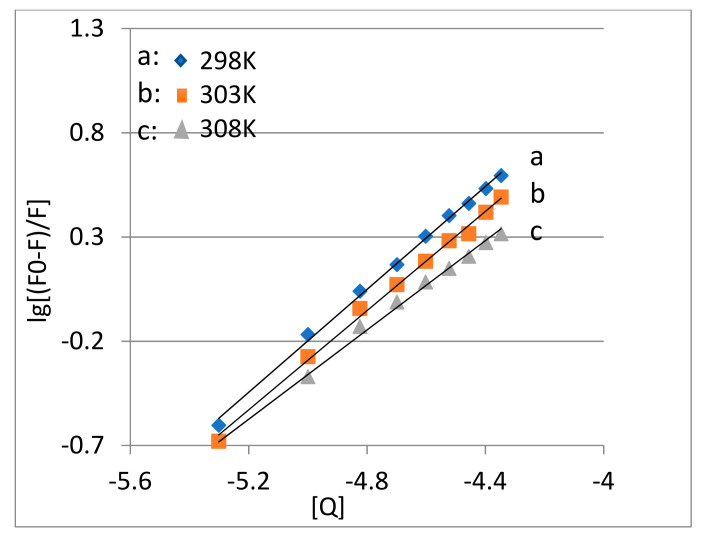
The Scatchard equation for the quenching of TRP by MC at different temperatures.

**Figure 4 molecules-28-02656-f004:**
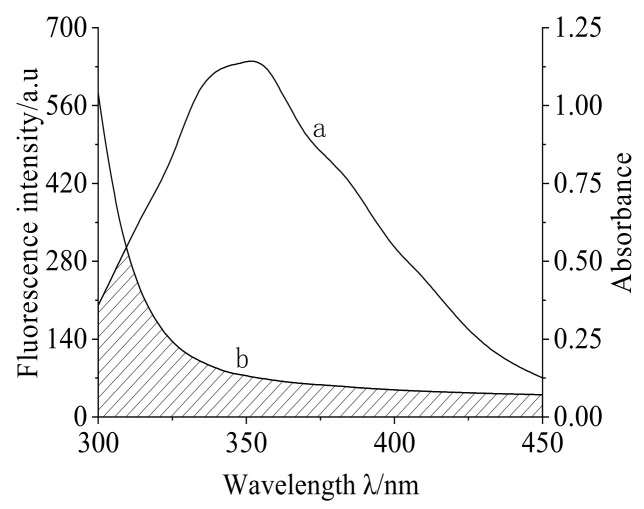
The overlapping spectra (a) of TRP fluorescence emission spectra and UV absorption spectra (b) of MC. C_(TRP)_ = 3 × 10^−6^ mol/L; C_(MC)_ = 3 × 10^−6^ mol/L; pH = 7.4; T = 298 K.

**Figure 5 molecules-28-02656-f005:**
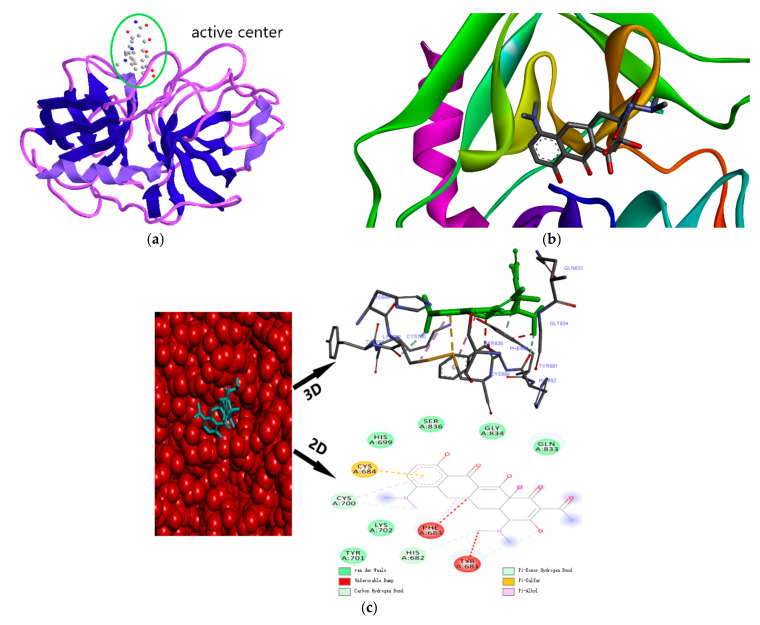
Molecular docking simulation of TRP and MC binding; (**a**) the molecular alignment diagram of MC and TRP; (**b**) the docking diagram of MC and TRP ribbon-like models; (**c**) the micro-environment 2D, 3D schematic diagram of MC combined with TRP.

**Figure 6 molecules-28-02656-f006:**
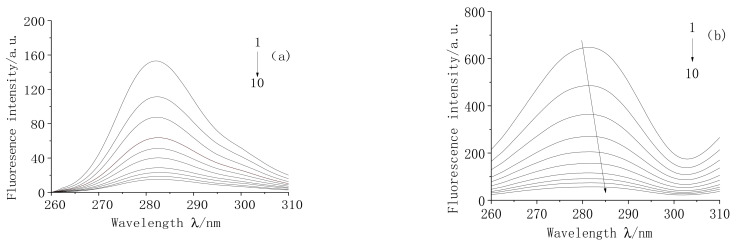
Synchronous fluorescence spectra of MC and TRP. (**a**):∆λ= 15, (**b**): ∆λ = 60; T = 298 K; pH = 7.4; C_(TRP)_ = 3 × 10^−5^ mol/L; C_(MC)_ (1–10) 0, 0.5, 1, 1.5, 2, 2.5, 3, 3.5, 4, 4.5 × 10^−4^ mol/L.

**Figure 7 molecules-28-02656-f007:**
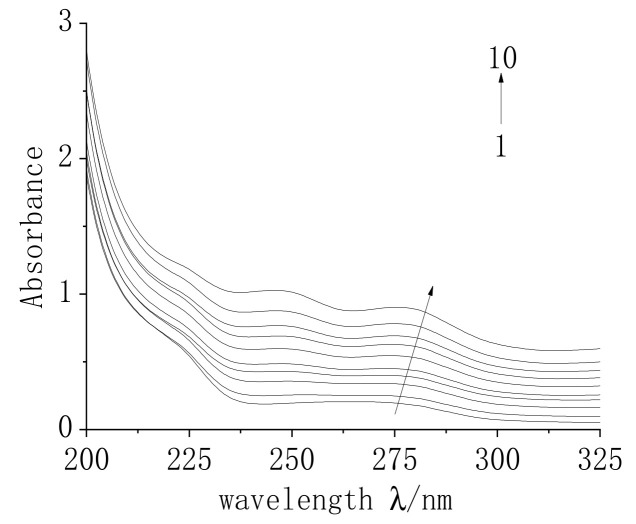
Ultraviolet spectra of MC and TRP. C_(TRP)_ = 1 × 10^−4^ mol/L; C_(MC)_= (1–10) 0, 0.5, 1, 1.5, 2, 2.5, 3, 3.5, 4, 4.5 × 10^−5^ mol/L; pH = 7.4; T = 298 K.

**Figure 8 molecules-28-02656-f008:**
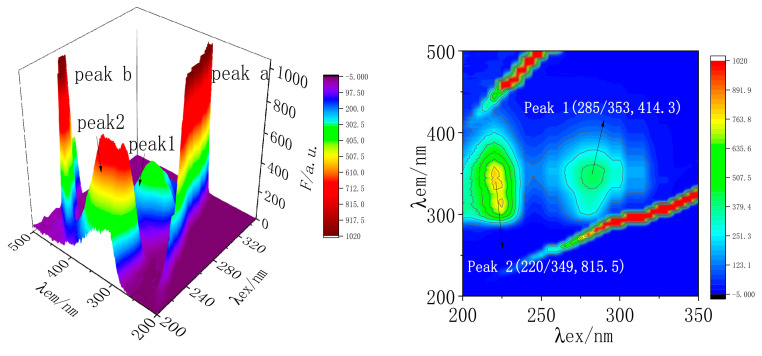
Three-dimensional fluorescence spectra and contour maps of TRP. C_(TRP)_ = 1.5 × 10^−7^ mol/L; T = 298 K; pH = 7.40.

**Figure 9 molecules-28-02656-f009:**
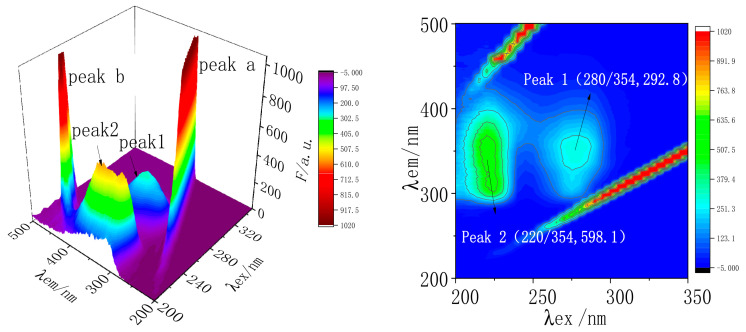
Three-dimensional fluorescence spectra and contour maps of MC-TRP. C_(TRP)_ = 1.5 × 10^−7^ mol/L; C_(MC)_ = 1 × 10^−5^ mol/L; T = 298 K; pH = 7.40.

**Figure 10 molecules-28-02656-f010:**
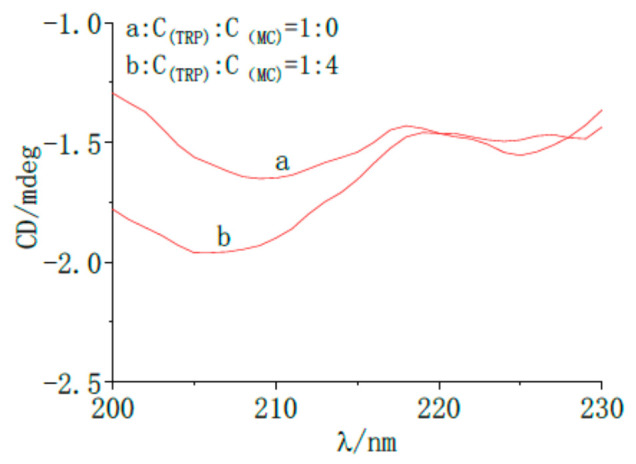
The circular dichroism spectra of MC interacting with TRP. (a): C_(TRP)_ = 1 × 10^−4^ mol/L; (b): C_(TRP)_ = 1 × 10^−4^ mol/L, C_(MC)_ = 1 × 10^−4^ mol/L.

**Table 1 molecules-28-02656-t001:** Quenching constants for the interaction of TRP and MC at different temperatures.

**T/K**	**Linear Equation**	**Correlation Coefficient *R^2^***	***K_sv_*/(L/mol)**	***K_q_*/(L/mol·s)**
298	y = 1.643x + 0.147	0.9920	1.643 × 10^5^	1.643 × 10^13^
303	y = 1.532x + 0.034	0.9917	1.532 × 10^5^	1.532 × 10^13^
308	y = 1.442x + 0.018	0.9933	1.442 × 10^5^	1.442 × 10^13^

**Table 2 molecules-28-02656-t002:** Binding constants and thermodynamic parameters of TRP and MC under different temperatures.

**T/K**	**Linear Equation**	**Correlation Coefficient *R^2^***	***K_A_*/(L/mol)**	**Binding-Site Number *n***
298K	y = 1.234x + 5.972	0.9973	9.381 × 10^5^	1.234
303K	y = 1.189x + 5.655	0.9959	4.536 × 10^5^	1.189
308K	y = 1.071x + 4.995	0.9950	9.910 × 10^4^	1.071

**Table 3 molecules-28-02656-t003:** Thermodynamic parameters of MC-TRP system at different temperatures.

***T*/K**	***K_A_*/(L/mol)**	**Δ*H*/(KJ/mol)**	**Δ*S*/(J/(mol.k))**	**Δ*G*/(KJ/mol)**
298	9.381 × 10^5^			−34.07
303	4.536 × 10^5^	−109.31	−252.48	−32.81
308	9.910 × 10^4^			−29.46

**Table 4 molecules-28-02656-t004:** Three-dimensional fluorescence spectra characteristic of TRP and MC-TRP.

**System**	**Component**	**Peak 1 (*λ_ex_/λ_em_, F*)**	**Peak 2 (*λ_ex_/λ_em_, F*)**
A	TRP	285/353, 414.3	220/349, 815.5
B	MC-TRP	280/354, 292.8	220/354, 598.1

**Table 5 molecules-28-02656-t005:** Secondary Structure Content of MC before and after TRP.

**System**	**α-Helix (%)**	**β-Sheet (%)**	**β-Turn (%)**	**Bandom Coil (%)**
TRP	10.8	50.5	19.2	27
TRP-MC	12.6	53.2	18.9	25.8

## Data Availability

Not applicable.

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
