# Peer review of "Characterization of the Interactions between Minocycline Hydrochloride and Trypsin with Spectroscopic and Molecular Docking Technology"

_molecules, 2023, doi:10.3390/molecules28062656_

Round 1

Author Response

The manuscript has been carefully edited by native English speaking colleagues. Please see the attachment.

Reviewer 2 Report

The authors in this manuscript presented various techniques to interpret the interaction of minocycline hydrochloride and trypsin. This manuscript is very well written, but it may be of interest only to a small group of researchers. I think this manuscript can be accepted for publication, but prior to this process the authors should add conclusions as to whether minocycline is trypsin toxic. This was the hypothesis of this manuscript.

Reviewer 3 Report

 I read the manuscript entitled “Characterization of the interactions between Minocycline Hy-drochloride and Trypsin with Spectroscopic and Molecular Docking technology” and have the following suggestions and comments to improve the state of the study.

The paper is not in good format (The work is on Template from 2022.)

In subsection 2.2 (page 2) the authors wrote the excitation wavelength was 280nm, the slit width was 7 nm, the scanning speed was 1500 nm/min, and the fluorescence emission spectrum was recorded in the wavelength range of 290nm-550nm. “

The same text is repeated in subsection 2.4 “ The excitation wavelength of the fluo-rescence spectrophotometer was 280 nm, the width of the slit was 7 nm, the scanning speed of the spectrum was 1500 nm/min, and the emission spectrum of the emission wave-length was from 290nm to 550nm.” Please check and correct

Page 4: where ?0 appears in equation 1(why it is explained under the equation)?

Page 4: The authors need to clarify who is the quencher in the studied system. If it is MC then they need to rewrite "TRP in MC" from the line after equation 1.

Page 4: The following text should be rewritten: “The Stern-Volmer curve of MC-TRP at three temperatures (298K, 303K, 308K) was plotted with the concentration of F0/F to the MC [Q]. The Stern-Volmer curve is shown in Fig. 2. “

Page 5: the following lines say the same thing, repetitive, please check and correct “The slope of the fitted line was the number of binding sites n, the intercept value between the line and the vertical coordinate was lg??. The number of bits n is calculated by the slope of the curve, and the constant KA is calculated according to the intercept of the curve. “

Page 5: where is fig 3, is missing?

please check equations 6-8, Equation 6 starts with an equal sign. Where it appears in equation 7 K2 maybe with superscript (in explanation)

Fig 5c put better resolution.

Minor points: Page 2: between drug small molecules =between small drug molecules; Page 2: investigate of binding interaction =investigate binding interaction; Page 6: ΔH, ΔS, and ΔG represents = ΔH, ΔS, and ΔG represent;  Table 3: different temperature. = different temperatures. Page 8 :the instrument are set to a fixed value. = the instrument is set to a fixed value.

Author Response

(The authors gave the same response as above.)

Round 2

Reviewer 3 Report

The authors have considered my comments. This paper could be accepted for publication.